# Application of a Biphasic Mathematical Model of Cancer Cell Drug Response for Formulating Potent and Synergistic Targeted Drug Combinations to Triple Negative Breast Cancer Cells

**DOI:** 10.3390/cancers12051087

**Published:** 2020-04-27

**Authors:** Jinyan Shen, Li Li, Niall G. Howlett, Paul S. Cohen, Gongqin Sun

**Affiliations:** 1Department of Cell and Molecular Biology, University of Rhode Island, Kingston, RI 02881, USA; 2Department of Biochemistry and Molecular Biology, Shanxi Medical University, Taiyuan 030001, China; 3Department of Cell Biology and Medical Genetics, Shanxi Medical University, Taiyuan 030001, China

**Keywords:** biphasic inhibition, combination targeted therapy, MDA-MB-231, MDA-MB-468, protein kinase inhibitors, triple negative breast cancer

## Abstract

Triple negative breast cancer is a collection of heterogeneous breast cancers that are immunohistochemically negative for estrogen receptor, progesterone receptor, and ErbB2 (due to deletion or lack of amplification). No dominant proliferative driver has been identified for this type of cancer, and effective targeted therapy is lacking. In this study, we hypothesized that triple negative breast cancer cells are multi-driver cancer cells, and evaluated a biphasic mathematical model for identifying potent and synergistic drug combinations for multi-driver cancer cells. The responses of two triple negative breast cancer cell lines, MDA-MB-231 and MDA-MB-468, to a panel of targeted therapy drugs were determined over a broad range of concentrations. The analyses of the drug responses by the biphasic mathematical model revealed that both cell lines were indeed dependent on multiple drivers, and inhibitors of individual drivers caused a biphasic response: a target-specific partial inhibition at low nM concentrations, and an off-target toxicity at μM concentrations. We further demonstrated that combinations of drugs, targeting each driver, cause potent, synergistic, and cell-specific cell killing. Immunoblotting analysis of the effects of the individual drugs and drug combinations on the signaling pathways supports the above conclusion. These results support a multi-driver proliferation hypothesis for these triple negative breast cancer cells, and demonstrate the applicability of the biphasic mathematical model for identifying effective and synergistic targeted drug combinations for triple negative breast cancer cells.

## 1. Introduction

According to the American Cancer Society, breast cancer will account for 30% of all female cancers, and is the second leading cause of cancer deaths in women in 2020 [1]. Early diagnosis and advances in targeted therapy have significantly improved the effectiveness of breast cancer treatment [2]. Targeted therapy is premised on blocking the dominant proliferative drivers of a given cancer, and it has translated into two major forms of treatment for breast cancers: (1) hormonal therapy for breast cancers that display receptors for estrogen and/or progesterone [3]; and (2) targeted therapy using monoclonal antibodies and small molecule inhibitors against ErbB2 for cancers that over-express this receptor protein tyrosine kinase [4]. 

Triple negative breast cancers (TNBC) are a group of heterogeneous cancers characterized by their lack of estrogen receptor (ER), progesterone receptor (PR), and ErbB2/HER2/Neu [5]. No dominant actionable targets have been identified and no targeted therapy has been approved for TNBC [6]. Consequently, TNBC still relies on surgery, radiation, and chemotherapy [5]. While TNBC represents only about 15–20% of all breast cancers, it is more aggressive with higher rates of recurrence and shorter overall survival [4]. There is an urgent need to find approaches of targeted therapy for TNBC [5,7].

While TNBC’s are readily defined by their lack of ER, PR, and ErbB2, it is much more difficult to define what they have in common at the molecular level due to their notorious inter-tumor heterogeneity. A comprehensive analysis of the molecular characteristics of TNBC, including genomic DNA copy number arrays, DNA methylation, exome sequencing, messenger RNA arrays, microRNA sequencing, and reverse-phase protein arrays, revealed that TNBCs are commonly associated with alterations in TP53, RB1, and BRCA1 function [8]; however, no dominant signaling protein was found to be commonly affected. *PIK3CA* was the most commonly mutated signaling gene at 9%, even though the PI(3)K pathway activity was affected more frequently by other alterations such as loss of *PTEN* and *INPP4B* and/or *PIK3CA* [8]. Blocking Akt, a central step in the PI3-kinase pathway has not proved to be an effective therapy [9]. Drugs for many other targets have been tested, including BRCA1/2, CDKs, receptor tyrosine kinases, angiogenesis (via vascular endothelial growth factor receptor), Src, and WNT signaling. Many clinical trials have tested combinations of targeted therapeutics or combinations with chemotherapy [6]. Despite these efforts, no effective targeted therapy for TNBC has emerged. 

At the center of targeted cancer, drug discovery is the analysis of how cancer cells respond to treatment by various drugs. Historically, the analysis of how cancer cells respond to treatments has used various versions of the Hill equation [10], which was originally developed to describe how O_2_ binds to hemoglobin [11]. When applied to cell responses to drug treatment, the full Hill equation (I = I_max_ × D^n^/(IC_50_*^n^ + D^n^)) uses three parameters to describe the response of biological systems to pharmaceutical intervention: I_max_ (maximal inhibition at saturating drug concentration), n (Hill co-efficient), and IC_50_*, the concentration of a drug that achieves 50% of the I_max_ [12]. 

When applied to how colorectal cancer cells responded to kinase inhibitors [12], the Hill co-efficient, n, varied between 0.3 and 0.8 suggesting varying levels of negative cooperativity. However, there is no obvious mechanistic explanation for this negative cooperativity. Furthermore, in some cases, the dose response curves were clearly broken into two phases, suggesting that a targeted drug may inhibit cell viability by interacting with two distinct targets with different affinities. Based on these considerations, we developed a biphasic mathematical model for characterizing the cell responses to targeted therapy [12]. The biphasic model assumes two inhibitory effects, and breaks the inhibition of a cancer cell by a targeted drug into a target-specific inhibition (F_1_ with K_d1_) and an off-target inhibition (F_2_ with K_d2_). In this model, the inhibition of cell viability by a drug as a function of drug concentration (D) follows this equation: I = F_1_ × [D]/([D] + K_d1_) + F_2_ × [D]/([D] + K_d2_). We further demonstrated that the biphasic inhibition only applies to multi-driver cancer cells, and toward mono-driver cancer cells, the inhibition becomes monophasic, with F_2_ inhibition becoming negligible. Thus, the biphasic model was able to distinguish multi-driver from mono-driver cancer cells. Furthermore, by identifying inhibitors for each driver, and quantifying the amplitude (F_1_) and the potency (K_d1_) of the inhibition by blocking each driver, the biphasic analysis was able to suggest potent and synergistic combinations for blocking colorectal cancer cells [12]. 

In light of the challenge of developing targeted therapy for TNBC, and their apparent multi-driver nature, we tested if the biphasic mathematical model is applicable to TNBC cells, and can identify potent and synergistic combinations of targeted therapy. The results indicated that the multi-driver hypothesis, biphasic analysis, and mechanism-based combination targeted therapy are directly applicable to MDA-MB-231 and MDA-MB-468, raising the prospect of developing targeted combination therapies for TNBC. 

## 2. Results

### 2.1. Profiling of MDA-MB-231 and MDA-MB-468 Responses to Kinase Inhibitors

To examine if the multi-driver proliferation hypothesis and the biphasic mathematical model apply to TNBC cells, we examined two TNBC cell lines, MDA-MB-231 and MDA-MB-468. Both cell lines have been widely used for studying the molecular mechanisms of TNBC proliferation and for drug discovery [13]. Both are included in the NCI-60 cell line panel, and widely used for cancer cell drug screening [14]. 

To determine the response of MDA-MB-231 and MDA-MB-468 cells to targeted therapy drugs, they were screened against a panel of 18 inhibitors against many common oncogenic protein kinases (Table 1). To gain a full assessment of the responses of these cells, the inhibitors were tested at 16 concentrations from 0.6 nM to 20 μM. The most potent inhibitor for MDA-MB-231 is the Src/Abl/PDGFR inhibitor dasatinib with an IC_50_ of 0.578 ± 0.05 μM, and the most potent inhibitor for MDA-MB-468 is the ErbB2/EGFR inhibitor lapatinib with an IC_50_ of 0.3 ± 0.06 μM. Both cell lines responded to a few other inhibitors, but with IC_50′_s above 1 μM. 

### 2.2. The Inhibition of MDA-MB-231 by Dasatinib Is Biphasic

To gain insights into the inhibition, the responses of MDA-MB-231 cell viability by dasatinib was analyzed by both the Hill equation and the biphasic model (Figure 1A). On the left is the inhibition data fitted to the Hill equation: I = I_max_ × D^n^/(IC_50_*^n^ + D^n^), where I_max_ is the maximal inhibition by dasatinib, IC_50_* is the IC_50_ for inhibiting the portion of cell viability that is sensitive to dasatinib, and n is the Hill coefficient, or slope. The data fit the equation reasonably well, with a root mean square error (RMSE) of 0.05. The curve fitting yielded an IC_50_* 0f 0.3 μM, and I_max_ of 88%, and an n of 0.49. The curve fitting had two issues. First, the data had a clear biphasic contour that was not reflected in the fitted curve, resulting in stretches of the data either above or below the curve. Second, the n of 0.49 suggests a strong negative cooperativity in dasatinib inhibition. It is not clear what mechanism this apparent negative cooperativity represents. When the data was curve-fitted to the biphasic equation: I = F_1_ × [D]/([D] + K_d1_) + F_2_ × [D]/([D] + K_d2_), a significantly better fit was obtained with an RMSE of 0.018. The biphasic fitting yielded three inhibitory parameters: an F_1_/F_2_ ratio of 49/51, and a K_d1_ of 29.9 nM, and a K_d2_ of 9.3 μM. These data suggested that dasatinib inhibited MDA-MB-231 in two phases: the first phase (F_1_) accounted for 49% cell viability, and had a K_d_ of 29.9 nM, and the second phase accounted for 51% of cell viability, and had a K_d_ of 9.3 μM. Because dasatinib blocks its molecular targets, such as Src, Abl, and PDGFR family kinases, with K_d_’s in the low nM range, the F_1_ inhibition is taken as the target-specific inhibition by dasatinib, while the F_2_ inhibition (K_d2_ of 9.3 μM) likely reflected an off-target toxicity. The interpretation suggests that the dasatinib target is responsible for a fraction (49%) of cell viability, consistent with the multi-driver hypothesis. A visual inspection of the two graphs (Figure 1A) also clearly supports the biphasic interpretation.

The main proliferative targets for dasatinib include Src, Abl, and PDGFR family kinase [15,16,17]. Because the cells did not respond to Abl inhibitor, imatinib (Appendix A for Table 1), or PDGFR inhibitor, sunitinib, the most likely target for dasatinib is Src family kinases. Previous reports indicate that Src does play a significant role in MDA-MB-231 proliferation [18,19,20]. The fact that inhibiting this target, likely Src, only blocks 49% of cell proliferation indicates that additional dasatinib-insensitive signaling pathway(s) is independently promoting viability when the dasatinib target is blocked. This prompted us to examine the responses of MDA-MB-231 to other kinase inhibitors in search of additional signaling pathways.

An examination of the inhibition curves by other kinase inhibitors indicated that even though AZD-6244, a Mek inhibitor, did not reach 50% inhibition at 20 μM, it did cause a partial inhibition of MDA-MB-231 viability even at nM concentrations (Figure 1B). In comparison, although sunitinib reached 50% between 10 and 20 μM, it did not inhibit MDA-MB-231 viability below 1 μM. In contrast, AZD-6244 caused a partial but consistent inhibition even at nM concentrations. The AZD-6244 inhibition data fit the Hill equation well (Figure 1C, left) (RMSE = 0.021), with an I_max_ of 46%, IC_50_* of 209 nM and an n of 0.68. The data fit the biphasic equation similarly well with an RMSE of 0.023 (Figure 1C, right). The biphasic analysis revealed that AZD-6244 inhibited 36% of viability (F_1_) with a K_d1_ of 91 nM, with the remaining portion of proliferation largely resistant to AZD-6244, with the K_d2_ of over 100 μM. Both the Hill analysis and the biphasic analysis support the following general conclusion: AZD-6244 partially blocks MDA-MB-231 viability, suggesting that the target, Mek, plays a partial but important role in the viability of the cell line. Both analyses indicated that AZD-6244 caused relatively little off-target inhibition. The discrepancy between the two analyses is minor: the Hill equation assumed a moderate negative cooperativity, while the biphasic model assumed mixed inhibition. Because the second phase inhibition is minor due to the high specificity of AZD-6244, the two analyses largely agreed with each other. The high specificity of AZD-6244 suggested by this data is consistent with the kinome scan data (http://lincs.hms.harvard.edu/db/datasets/20156/results) that indicated that AZD-6244 binds to Mek 1 and Mek 2 with K_d_’s of 99 and 530 nM, respectively. It did not bind to any other kinase with a K_d_ below 1 μM. 

### 2.3. MDA-MB-231 Viability Is Potently Inhibited by the Combination of Dasatinib and AZD-6244

The biphasic analysis indicated that 49% of MDA-MB-231 viability can be blocked by dasatinib with a K_d_ of 29.9 nM, while 36% of viability can be blocked by AZD-6244 with a K_d_ of 91 nM. The likely targets for dasatinib, Src, and AZD-6244 target, Mek, represent distinct signaling pathways, thus the inhibitory effects of these two drugs should combine to achieve much stronger inhibition than either inhibitor alone. This prediction was tested (Figure 2A) by measuring the cell viability in the presence of both drugs at equal concentrations. The combination of dasatinib and AZD-6244 was indeed dramatically more effective in blocking MDA-MB-231 viability than either drug alone. The two-drug combination had IC_50_ and IC_70_ of 73 nM and 158 nM, respectively, much lower than that of the most sensitive individual inhibitor, dasatinib, with IC_50_ and IC_70_ at 697 nM and 7413 nM, respectively. The dramatic synergy is illustrated by the isobolograms in Figure 2B,C, as the dots representing the combined IC_50_ or IC_70_ are far below the line linking the individual IC_50_’s or IC_70_’s. The dose reduction index could not be meaningfully calculated because AZD-6244 only reached about 40% inhibition. It is also noteworthy that the synergy is most dramatic at high inhibition levels, such as 80% or 90% inhibition. This feature is mathematically consistent with the multi-driver proliferation mechanism, and may be an especially desirable feature for targeted therapy. Finally, while dasatinib did reach 80% and 90% inhibition, this level is achieved by both target-specific and off-target effects. The off-target effects represented by the F_2_ inhibition, while artificially helpful in enhancing the in vitro inhibitory effect, is unlikely to contribute to effective targeted therapy. It is likely the cause of dose-limiting toxicity. In contrast, the drug combination reached 90% inhibition at about 600 nM of both drugs. Based on the K_d2_ values of both drugs, the drug combination at this concentration would not be expected to cause general toxicity. 

Overall, these results clearly support the multi-driver hypothesis for this cancer cell line in that the cell viability is dependent on both Src kinases and the MAP kinase pathways. Inhibiting either pathway alone is not sufficient to effectively block these cells, while the drug combination can effectively block cell viability with strong synergy. 

### 2.4. MDA-MB-468 Cells Follow Similar Inhibitory Patterns but Are Sensitive to Different Drugs

To determine if the multi-driver proliferation mechanism and the biphasic mathematical model also apply to other TNBC cancer cells, the same analytical approach was applied to the cell line MDA-MB-468. The most potent inhibitor for MDA-MB-468 is lapatinib, with an IC_50_ of 231 nM (Table 1). Fitting the MDA-MB-468 data to the Hill equation generated a well-fitting curve (RMSE = 0.026) with n of 0.45, IC_50_* of 0.19 μM, and I_max_ of 100% (Figure 3A, left). The n value of significantly less than 1 suggested an apparently strong negative cooperativity consistent with the flattened inhibition curve. Biphasic analysis of the same data generated a similarly well-fitting curve (RMSE = 0.038), but with a very different mechanistic interpretation for the flattened inhibition pattern. It suggested that lapatinib inhibited 53% (F_1_) of MDA-MB-468 viability with a K_d1_ of 17 nM, and inhibited the remaining 47% with a K_d2_ of 3.05 μM. The relatively close K_d1_ and K_d2_ (a difference of 179 fold) fused the two phases together to make the biphasic nature of the dose response curve less visually obvious. Lapatinib is an EGFR and ErbB2 inhibitor and MDA-MB-468 is negative for ErbB2 but overexpresses EGFR [21]. The sensitivity of MDA-MB-468 to lapatinib has been previously reported [22]. However, the biphasic analysis suggests that lapatinib only achieves a partial inhibition through a target-specific inhibition, and achieves the remaining inhibition by an off-target effect. This result suggests that the lapatinib target, presumably EGFR, only supports 53% of MDA-MB-468 viability, leaving the remaining portion of viability dependent on other driver(s). 

The analysis of MDA-MB-468 responses to targeted drugs by the Hill equation and the biphasic equation offered an interesting comparison of the two models. Both models fit the data similarly well statistically, with RMSE of 0.026 for the Hill equation and 0.038 for the biphasic equation. The Hill equation assumed a single target binding, and relied on an assumed strong negative cooperativity (n = 0.45) to generate a well-fitting curve, generating an I_max_ of 100% and an IC_50_ of 190 nM. In contrast, the biphasic model assumed a biphasic inhibition pattern, relied on varying the F_1_/F_2_ ratio (53/47), K_d1_ (17 nM), and K_d2_ (3.05 μM) to generate a well-fitting curve. Clearly, the two models generated two dramatically different interpretations of the response data. The Hill analysis suggests that lapatinib binds to a single molecular target with negative cooperativity to cause all the observed inhibition. In contrast, the biphasic equation suggests that lapatinib causes 53% of viability inhibition by binding to one target with a K_d_ of 17 nM, and inhibits the remaining 47% by an additional mechanism (K_d_ of 3.05 μM). Statistics alone cannot distinguish these mechanistic explanations, thus other information is necessary to distinguish these possibilities.

If the biphasic analysis is correct, then MDA-MB-468 should have additional drivers other than EGFR. To search for additional signaling pathways that would complement EGFR in supporting MDA-MB-468 viability, we examined the inhibition profiles of other kinase inhibitors. Even though GSK690693 was a weak inhibitor of MDA-MB-468 as indicated by the IC_50_ of 9.3 μM, we noticed that it partially inhibited MDA-MB-468 viability even at low nM concentrations (Figure 3B). Even though crizotinib and BMS-754807 both had lower IC_50_ values, GSK690693 clearly was more potent at the lower concentration range. The inhibition profile suggested that GSK690693 reached about 45% inhibition at approximately 1 μM and plateaued until 10 μM. The inhibitor caused additional inhibition at 20 μM, thus displaying a clearly biphasic inhibition pattern. Fitting the data to the Hill equation produced the graph in Figure 3C (left). The fit of the data to the Hill equation was reasonable as judged by the RMSE (0.08), and yielded an I_max_ of 100% and an IC_50_ of 2.94 μM, and again suggested an apparently strong negative cooperativity (*n* = 0.42). Furthermore, the Hill graph did not reflect the subtle but clear biphasic contour of the data. The data fit the biphasic equation slightly better (RMSE of 0.066) (Figure 3C, right), yielding an F_1_ of 37%, a K_d1_ of 67 nM, and a K_d2_ of 15.6 μM. GSK690693 is a specific Akt inhibitor [23], and this data suggested that Akt signaling accounted for 37% of MDA-MB-468 viability, and this portion of viability was blocked by GSK690693 with a K_d_ of 67 nM. 

We then determined how potent the combination of lapatinib and GSK690693 was toward MDA-MB-468. As shown in Figure 4A, the combination was indeed much more potent than either inhibitor alone. The combination had an IC_50_ and IC_70_ of 22 nM and 64 nM, respectively, dramatically lower than either drug individually. An inhibition of 90% was achieved below 1 μM of the drug combination. The dramatic synergy of the drug combination is illustrated by the dose reduction of 40-fold to achieve a 70% inhibition (Figure 4B). This result supports the biphasic interpretation and indicates that MDA-MB-468 is indeed a multi-driver cancer cell line, dependent on Akt and EGFR signaling, and that combination targeted therapy strategy would likely be an effective strategy for a cancer with a similar mechanism. 

### 2.5. The Effectiveness of the Drug Combinations Is Cell-Specific

Having identified an effective combination of dasatinib and AZD-6244 for MDA-MB-231, and lapatinib and GSK690693 for MDA-MB-468, we wondered if the effectiveness of the drug combinations is cell-specific. We therefore tested the effectiveness of dasatinib/AZD-6244 on MDA-MB-468 cells and the effectiveness of lapatinib/GSK690693 on MDA-MB-231 cells. The IC_50_ of dasatinib/AZD-6244 for MDA-MB-468 was 15 μM, more than 200 fold higher than that for MDA-MB-231 at 73 nM (Figure 5A). Furthermore, the drug combination had virtually no effect on MDA-MB-468 below 1 μM. The lapatinib/GSK690693 combination was similarly cell-specific. While it blocked MDA-MB-468 viability with an IC_50_ of 22 nM, it did not affect MDA-MB-231 viability up to 5 μM, and caused significant cell inhibition between 10 and 20 μM, a >500-fold difference. These data demonstrate that the potency of the drug combinations is strictly dependent on the drug targets playing crucial roles in a cell’s viability.

### 2.6. The Effects of Drugs and Their Combinations on the Signaling Pathways

The above results suggested that EGFR signaling and Akt signaling independently play crucial roles in MDA-MB-468 cells, while Src and the MAP kinase pathway play independent and crucial roles in MDA-MB-231 cells. Some of these responses are consistent with known genomic information on these cells. For example, MDA-MB-231 contains mutations in several genes in the MAP kinase pathway, such as EGFR, BRAF, IRS4, KRAS, and PDGFRA. Although most of these mutations have not been confirmed to be associated with oncogenic properties, it is known that G13D mutation is oncogenic by activating the MAP kinase pathway downstream to KRAS [24]. This explains the reliance of MDA-MB-231 on Mek signaling. Src kinases in MDA-MB-231 do not contain any mutations, but Src activation is rarely activated by mutations, but most frequently activated by increased expression or mis-regulation [25,26,27]. According to the COSMIC database, MDA-MB-468 overexpresses EGFR [28,29], which explains why this cell line is sensitive to lapatinib. Furthermore, it is well established that MDA-MB-468 cells are deficient in PTEN expression [30,31,32,33,34], which would lead to constitutive activation of Akt signaling. This observation explains the partial reliance of MDA-MB-468 on Akt activity indicated by the cell line sensitivity to GSK690693.

To determine the effects of the drug treatments on the signaling pathways, both cell lines were treated by the drugs individually and in combination for 2 h, and the protein levels and phosphorylation/activation status of numerous signaling proteins, including Src, and proteins from the MAP kinase and PI 3-kinase pathways were determined by Western blots (Figure 6). In MDA-MB-468 cells, lapatinib treatment mildly decreased Akt phosphorylation on T308, but not on S473, and near completely blocked Mek and Erk phosphorylation. These results suggested that lapatinib mainly blocked the MAP kinase pathway. GSK690693 is a specific Akt inhibitor, and it fully blocked the phosphorylation of PRAS40 and significantly decreased GSK-3β phosphorylation. Both PRAS40 and GSK-3β are direct Akt substrates [35]. The lapatinib and GSK690693 combination blocked the phosphorylation of PRAS40, GSK-3β, Mek, and Erk. Thus, the combination blocked both the Akt and MAP kinase signaling pathways. 

The effects of the lapatinib and GSK690693 on MDA-MB-231 cells provided an interesting contrast. First, lapatinib had no appreciable effect on the protein expression or phosphorylation levels of any of the proteins in MDA-MB-231. Second, the phosphorylation levels of Akt on both T308 and S473 are very low in untreated MDA-MB-231 cells. The Akt phosphorylation levels were unaffected by lapatinib, but greatly stimulated by GSK690693, and the lapatinib/GSK690693 combination. These results indicated that MDA-MB-231 cells did not rely on the EGFR kinase pathway or Akt signaling for viability, and thus were not sensitive to lapatinib and GSK690693, either individually or in combination. The observation that Akt phosphorylation on both T308 and S473 was increased by GSK690693 indicated that GSK690693 entered the cells, and interacted with Akt. Increased Akt phosphorylation in response to GSK690693 was previously observed in multiple cell lines [12,23,36,37,38]. The fact that GSK690693 and GSK690693/lapatinib combination did not inhibit the phosphorylation of PRAS40, and GSK-3β explains why MDA-MB-231 was not inhibited by GSK690693 or the combination. It is not yet clear why GSK690693 did not inhibit Akt phosphorylation of PRAS40 and GSK-3β in MDA-MB-231 cells. These results are consistent with literature reports that another Akt inhibitor, MK-2206 [39,40,41], also does not inhibit MDA-MB-231 cell viability, even though it inhibits Akt phosphorylation in MDA-MB-231 cells. These results indicated that EGFR and Akt play negligible roles in MDA-MB-231 proliferation. Collectively, these results provide a molecular basis for why lapatinib/GSK690693 combination blocks MDA-MB-468 viability specifically, but had no effect on MDA-MB-231.

The treatment of MDA-MB-231 cells with dasatinib decreased Src phosphorylation on both Y416 and Y527. Y416 is a site of autophosphorylation and Y527 is phosphorylated by a different PTK, Csk [42]. Both Src and Csk are sensitive to dasatinib [43]. Treatment of MDA-MB-231 with AZD-6244 resulted in a complete block of Erk phosphorylation by Mek. The combination of dasatinib and AZD-6244 resulted in decreased Src phosphorylation on both Y416 and Y527, and blocked phosphorylation of Erk. The fact that the drug combination effectively blocked MDA-MB-231 viability indicates that Erk and Src signaling is crucial for the viability of this cell line. Surprisingly, the treatment of MDA-MB-468 cells with dasatinib, AZD-6244, and the combination also caused the same inhibition pattern, even though the drugs did not block MDA-MB-468 viability. This result suggests that these pathways are present in MDA-MB-468 cells, but do not play appreciable roles in their viability. 

### 2.7. The Hill and Biphasic Analyses Applied to Multi-Driver versus Mono-Driver Cancer Cells

Up to this point, seven cell lines have been subjected to a detailed kinase inhibition profiling in our lab. The dose response data are analyzed by both the Hill equation and the biphasic equation, and the inhibitory parameters from these analyses are summarized in Table 2. The list contains two mono-driver cancers, an acute myeloid leukemia cell line, CTV-1 [44,45], non-small cell lung cancer cell line HCC-827 [46], three colorectal cancer cell lines, HT-29, SK-CO-1, and NCI-H747 [12], and two TNBC cell lines MDA-MB-231 and MDA-MB-468. CTV-1 and HCC-827 are both mono-driver cancer cells driven by mutated Lck kinase [47] and EGFR [28], respectively. The colorectal cancer cells and TNBC cells are multi-driver cancer cells. The combined Hill analysis and biphasic analysis offered clear and consistent distinguishing patterns for mono-driver and multi-driver cells. Mono-driver cancer cells displayed: (1) I_max_ and F_1_ close to 100%, (2) n values above 1, and (3) similarly low nM IC_50_* and K_d1_ values. In contrast, multi-driver cancer cells produced: (1) F_1_ values significantly below 100%, (2) n values significantly below 1, and (3) low nM K_d1_ values dramatically below the IC_50_*’s.

To determine if the multi-driver hypothesis, a biphasic inhibition pattern is broadly applicable to other triple negative breast cancer cells, we applied the Hill analysis and the biphasic analysis to the TNBC data in the Genomics of Drug Sensitivity in Cancers database (https://www.cancerrxgene.org/compound/Trametinib/1372/overview/ic50). The database contains 17 TNBC cell lines. Judging by the standard discussed above, only one cell line is a clear mono-driver cancer cell line, DU-4475. This cell line is sensitive to BRAF inhibitors AZ628, PLX472, and SB5908, to Mek inhibitors RDEA119, CI-1040, and PD-0325901, and Erk inhibitor VX-11e. Hill analysis and biphasic analysis (Table 3) indicate that the cell line responds to all seven inhibitors as a largely mono-driver cancer with the following characteristics. (1) The dose response data fit well to both the Hill equation and the biphasic equation; (2) The I_max_ and F_1_ are generally similar and close to 100%, indicating that all seven drugs can fully inhibit cell viability; (3) the IC_50_* from the Hill analysis and the K_d1_ from the biphasic analysis are in good agreement for every inhibitor; (4) the n values from the Hill analysis are consistently above 1. Indeed, DU-4475 cells contain a BRAF V600E mutation that would activate BRAF, which, in turn, activated Mek and Erk. Both the Hill analysis and biphasic analysis indicate that the DU-4475 cells can be fully blocked by inhibitors to BRAF, Mek or Erk, consistent with this being a mono-driver cell line. The other 16 TNBC cell lines in the GDSC database did not respond to kinase inhibitors strongly and consistently enough to suggest that any of them is a mono-driver cancer. Furthermore, the data are too incomplete (too few concentration points in a too narrow a range) and noisy (single determination with a single well for each concentration) to definitely show that they are multi-driver cancer cells. Cases in point are MDA-MB-231 and MDA-MB-468 cell lines. Both are among the 16 TNBC cell lines tested, and, in contrast to our data, the GDSC data do not demonstrate that they are multi-driver cancer cells. We therefore anticipate that most of the TNBC cells would turn out to be multi-driver cancer cells, and are subject to biphasic analysis and combination targeted therapy. 

## 3. Discussion

Targeted therapy is effective for mono-driver cancers, but not for most cancers. Genetic evidence indicates that most cancers are multi-driver cancers. Therefore, a combination of drugs, each blocking an independent driver, would be necessary for effective therapy. Most TNBC likely belongs to this category as no dominant driver has been found, and no targeted therapy has been approved. Identification of effective targeted therapy combination is a challenge since most current therapeutic combinations are formulated empirically and lack synergistic benefits [48]. Because most efforts to identify effective drug combinations rely on empirical screening [49,50,51,52], due to the large number of potential combinations, most screens test drug combinations only at very few concentrations. The development of a systematic and mechanism-based mathematical model for formulating drug combinations would greatly enhance the development of combination therapy for multi-driver cancers [53]. 

In a recent study, we developed a rational approach for identifying effective targeted therapy combinations for colorectal cancer cells [12]. The approach was based on the observation that multi-driver cancer cells have a biphasic response to targeted therapy drugs: a target-specific inhibition and an off-target inhibition. The traditional analytic tools, all based on the assumption of a single target binding, are inadequate for characterizing such a biphasic response, as has been suggested previously [54]. A biphasic mathematical model was developed that can accurately describe the biphasic response and determine the amplitude (F_1_ and F_2_) and the potency (K_d1_ and K_d2_) of the target-specific and off-target responses [12]. The previous study further demonstrated that combination of drugs with mechanistically distinct F_1_ effects resulted in highly potent and strongly synergistic combinations against colorectal cancer cells. In the current study, the same approach was applied to two triple negative breast cancer cell lines and the results suggest the multi-driver proliferation hypothesis, the biphasic mathematical model, and the mechanism-based combination targeted therapy strategy also apply to these triple negative breast cancer cells. 

### 3.1. Drug Sensitivity Profiling Combined with Biphasic Analysis Can Identify Partial Drivers and Suggest Effective Drug Combinations for TNBC Cells

Determining the oncogenic drivers for a given cancer is a challenge, and the TNBC cells are no exceptions. In this current study, we demonstrate that quantitative analysis of cancer cell response to kinase inhibitors can be an effective tool for identifying the drivers in triple negative breast cancer cells. First, the biphasic analysis offers a more comprehensive view of the signaling mechanism for the TNBC cells. MDA-MB-231 has been reported to respond to dasatinib [55,56,57,58,59] and AZD-6244 [60,61,62], MDA-MB-468 has been shown to respond to lapatinib [22,63,64] and AKT inhibitor MK-2206 [65] previously. The previous studies tend to view the cancer cell signaling with a focus on one driver or another, often in isolation from other signaling pathways. Examining the responses of a cancer cell to a panel of signaling inhibitors provides a more comprehensive view of the signaling landscape for a given cell, and separating the target-specific and off-target effects for a drug provides a more quantitative assessment of the role of a given kinase in cell viability. This analysis in the context of multi-driver proliferation offers a holistic view of cancer signaling mechanism. Second, assuming a multi-driver signaling mechanism and assessing the relative contribution of each driver to viability provide a mechanistic basis for formulating targeted drug combinations that can effectively and synergistically block cancer cell viability. This approach identified effective drug combinations for both MDA-MB-231 and MDA-MB-468 cells. In both cases, the drug combination is far more potent than the individual drugs alone. Finally, these results provide an explanation as to why mono-agent therapy would not be effective for TNBC. Even though both cells displayed some level of sensitivity to several kinase inhibitors, with sub-μM IC_50_’s, the relatively low IC_50_’s, achieved by the combined effect of target-specific and off-target inhibition, are deceiving. When used for targeted therapy, only the target-specific effect is likely beneficial for therapeutic efficacy, and the off-target effects are likely contributing to dose-limiting toxicity. If these drugs are used in the target-specific concentration window, the best they can achieve are approximately 50% inhibition. This provides at least one explanation for why single agent targeted therapy would not be effective for multi-driver cancers. Dasatinib [66], Akt inhibitor MK-2206 [9], and lapatinib [67] have all been tested against TNBC in clinical trials, and all displayed minimal benefits. In retrospect, unless a patient has a mono-driver cancer with one of these kinases as the driver, these mono-agent therapies would not be expected to be effective. 

Even though the target-specific effects are combined synergistically when appropriate drugs are used together, data in this study and a previous study [12] suggest that the off-target effects of the individual drugs are not combined synergistically. For example, dasatinib and AZD-6244 individually inhibited MDA-MB-468 with IC_50_’s of around 20 μM or above (Table 1), and the combination inhibited the cell line with IC_50_ close to 20 μM of each (Figure 5A). Similarly, lapatinib and GSK690693 inhibited MDA-MB-231 with IC_50_’s of >20 μM and 16.7 μM, respectively (Table 1), and the combination inhibited the cell line viability with an IC_50_ close to 10 μM of each (Figure 5B). Similar observations were made with drugs that inhibited HT-29 cells with only off-target effects [12]. A broader and unanswered question about this combination approach is whether these cell line-specific drug combinations would display any potent general toxicity toward any normal cells. Drug combinations should be tested against a broad spectrum of normal cells and/or animal models to assess their toxicity before they can be used in clinical setting.

### 3.2. The Hill Analysis versus Biphasic Analysis

Historically, the response of cells to drug treatment was performed using some version of the Hill equation. In this study and the preceding study, we have developed an alternative equation for analyzing the cell responses to targeted therapy. One fundamental difference between the Hill equation and the biphasic equation is the mechanistic assumption about the shallow drug response curves, a common feature of many drug response curves [54]. The Hill equation assumes that the shallow response curves are produced by a drug binding to a single target with negative cooperativity, producing n values less than 1. There is no mechanistic basis for this negative cooperativity. The Hill analysis data in Table 2 indicated that the shallow response curves (n < 1) are common among multi-driver cancer cells. The biphasic analysis interprets the shallow response curves as the result of an inhibitor binding to multiple targets, causing a partial inhibition in a target-specific manner and additional inhibition by an off-target effect. The competing interpretation is best illustrated by MDA-MB-468, where the I_max_ was 100% for both lapatinib and GSK690693 in the Hill analysis, while the F_1_ for the two drugs are 53% and 37%, respectively, according to the biphasic analysis. As a result, the IC_50_ values (2940 nM for GSK690693 and 190 nM for lapatinib) are significantly higher than the target-specific K_d1_ values (67 nM for GSK690693 and 17 nM for lapatinib). The remarkable synergy between lapatinib and GSK690693 (combined IC_50_ of 22 nM each) in inhibiting MDA-MB-468 viability strongly supports the biphasic interpretation. 

Theoretically, the inhibition pattern may contain more than two phases. For example, if a cell contains three drivers and each responsible for 33% cell viability, a drug that inhibits driver one with a K_d_ of 10 nM, the driver two with a K_d_ of 100 nM, and causes off-target toxicity with a K_d_ of 5 μM, then the inhibition will be theoretically triphasic. Such perfectly configured scenarios are likely rare. In practice, if two drivers both respond to the same inhibitor with K_d_’s within 10 fold of each other, they become indistinguishable by this type of analysis. The toxicity phase is also likely complex as it is caused by ill-defined and likely variable mechanisms. Despite these potentially complicating scenarios, the inhibition of most multi-driver cancer cells by most inhibitors for one of the drivers can be practically described by this biphasic mathematical model.

Even though the biphasic analysis assumed a bi-target inhibition, it is still informative in analyzing mono-driver cancers, where the F_2_ becomes negligible, and the equation becomes a simplified Hill equation with assumed n value of 1. For all the mono-driver cancer cell/drug combination, both the Hill analysis and the biphasic analysis produced similar inhibitory potency (IC_50_* vs. K_d1_). The only major difference is the apparent positive cooperativity suggested by the n values (>1) in the Hill analysis. The mechanistic basis for the positive cooperativity is still unclear.

### 3.3. Heterogeneity and Cell-Specificity of TNBC Signaling 

It is also important to note the dramatic mechanistic differences between the two cell lines revealed by both inhibitor sensitivity analysis and immunoblot analysis. MDA-MB-468 relies on the lapatinib-sensitive MAP kinase pathway and Akt signaling pathways for viability, while MDA-MB-231 relies on a KRAS-initiated downstream signaling and Src kinase for viability. Consequently, the potency of the drug combinations is extremely cell-specific. Dasatinib/AZD-6244 inhibited MDA-MB-231 >200-fold more potently than MDA-MB-468, while lapatinib/GSK690693 was >500-fold more potent toward MDA-MB-468. The cell specificity is also consistent with available genetic information. MDA-MB-231 contains KRAS G13D mutation and the cell was partially sensitive to AZD-6244, while MDA-MB-468 does not contain the mutation and was not sensitive to the inhibitor. Lapatinib inhibited the phosphorylation of both Mek and Erk phosphorylation in MDA-MB-468, but not in MDA-MB-231, indicating that the MAP kinase pathway activation is dependent on EGFR in MDA-MB-468 cells but not in MDA-MB-231. The roles of Akt in the two cell lines are also quite different. GSK-690693 blocked Akt phosphorylation of its substrates PRAS40 and GSK-3β in MDA-MB-468 cells, but not in MDA-MB-231 cells. This was not due to a lack of cell uptake or cellular export of the drug, as GSK690693 strongly stimulated the phosphorylation of Akt on both T308 and S473 in MDA-MB-231 cells. These results demonstrate drug sensitivity is closely linked to the cell signaling mechanism, rather than general cytotoxicity. This feature is likely important for the future application of combination therapy in clinical settings. 

### 3.4. Implications of Biphasic Analysis on Targeted Therapy

Identifying synergistic targeted therapy is still a major challenge for cancer treatment. The strategy demonstrated in this paper and the previous study [12] provides an intriguing strategy for this purpose. To fully evaluate the application of this strategy, two types of further investigation will be necessary. First, the combinations that were found to be effective in vitro need to be tested in vivo for their effectiveness in inhibiting tumor growth in animals. Second, the biphasic analysis has been applied only to a small number of cell lines and needs to be tested in more cell lines representing diverse cancer types. Based on the colorectal cancer and TNBC cell lines tested thus far, the Src kinase, the MAP kinase pathway, and PI-3 kinase/Akt pathways play crucial roles as cancer drivers. It is likely that, as more cell lines are tested, mechanistic patterns will emerge and cancer cells will be classified based on their molecular signaling mechanism and drug sensitivity. It is also possible that relatively few combinations of drugs will be found to be effective for multiple types of cancers. 

## 4. Materials and Methods 

### 4.1. Cell Lines, Media, and Drugs

The TNBC cell lines, MDA-MB-231 and MDA-MB-468, were purchased from ATCC (Manassas, VA, USA). They were authenticated by ATCC by short tandem repeat profiling, cell morphology monitoring, karyotyping, and cytochrome C oxidase I testing. The cells were grown in ATCC-recommended media, containing 10% FBS and 1% Penicillin-streptomycin (Thermo Fisher Scientific, Pittsburgh, PA, USA). All cells were cultured at 37 ℃ in humid atmosphere containing 5% CO_2_. Kinase inhibitors were purchased from Selleckchem (Munich, Germany), LC Laboratories (Woburn, MA, USA), or AdooQ Bioscience (Irvine, CA, USA). 

### 4.2. Cell Culture and Viability Assays

Cell culture and viability assays were performed as described previously [12]. Briefly, cells were plated in 96-well plates at 25,000 cells per well, and were incubated with drugs at indicated concentrations in 100 μL medium containing 1% DMSO for 72 h. At the end of the incubation, cell viability was determined by the MTT dye assay (Thermo Fisher Scientific). The formazon product was determined by absorbance at 490 and 750 nm using a VersaMax Tunable Microplate Reader. The A490-A750 values were taken as indicator of cell viability. All cell growth and drug inhibition experiments were performed in triplicates at least two times. 

### 4.3. Curve-Fitting by the Hill Equation and the Biphasic Equation

Curve fitting was performed in Microsoft Excel. The Hill equation used was: I = I_max_ × D^n^/(IC_50_*^n^ + D^n^), where I_max_ is the maximal inhibition by a drug, IC_50_* is the IC_50_ for inhibiting the portion of cell viability that is sensitive to the drug, and n is the Hill coefficient, or slope. In curve fitting, the root mean square error (RMSE) between the experimental cell viability and the calculated viability was minimized using the I_max_, IC_50_*, and n as a variable. The RMSE minimization was carried out using the Solver add-in program in Microsoft Excel. 

The biphasic kinetic curve fitting was performed similarly, except that the equation used was: I = F_1_ × [D]/([D] + K_d1_) + F_2_ × [D]/([D] + K_d2_). In this equation, the inhibition of cell viability (I) as a function of variable drug concentration is determined by three constants: F_1_, K_d1_, and K_d2_. The F_2_ was not an independent variable, but calculate from F_1_ (F_2_ = 100%-F_1_). The RMSE was minimized allowing the F_1_, K_d1_, and K_d2_ as variables. The F_1_, K_d1_, and K_d2_ that resulted in the smallest RMSE were taken as the best fitting parameters. 

### 4.4. Drug Synergy Analysis and Combination Index Calculation

Drug synergy was evaluated by two methods: quantitatively by the dose reduction index (DRI), and visually by the isobologram, as described previously [12]. Cell viability was determined after incubation in the presence of each drug alone or both drugs (1:1 ratio) at 16 concentrations ranging from 0.6 nM to 20 μM. The concentration of a drug or drug combination that inhibited cell viability at certain percentage (IC_x_) was determined manually from the graphs of drug dose response curves. The DRI was calculated according to Chou [68], using the Chou and Talalay Method [68]. The DRI at a given level of inhibition was calculated by the following equation: DRI = IC_x−1_ × IC_x−2_/(IC_x−1+2_ x (IC_x−1_ + IC_x−2_), where IC_x−1_, IC_x−2_, and IC_x−1+2_ are the concentrations of drug 1, drug 2, and drugs 1 + 2, causing X% inhibition of cell viability, respectively. Isobolograms, comparing the inhibitory potency of a drug combination to individual drugs, was drawn using Microsoft Excel.

### 4.5. Western Blot Analysis of Drug Effects on Cell Signaling

To determine the effects of the kinase inhibitors on the signaling network in the TNBC cells, the cells were cultured to 60–70% confluency, and treated by the drugs or drug combinations at indicated concentrations for 2 h under normal cell culture conditions. After treatment, the culture medium was aspirated, and adherent cells were then resuspended in the SDS-PAGE loading buffer containing a protease inhibitor cocktail (Sigma-Aldrich, St. Louis, MO, USA) and protein phosphatase inhibitors (PhosSTOP, Sigmal-Aldrich) and lysed in a 95 °C heat bath immediately. SDS-PAGE and Western blotting were performed as described [69]. The loading was normalized based on the β-actin expression levels in the lysates determined by an antibody specific for this protein. Western blotting analyses were performed on two separate experiments. All antibodies were purchased from Cell Signaling Technology (Danvers, MA, USA). The density of each protein band in the Western blots was measured using the ImageJ software (https://imagej.nih.gov/ij/).

### 4.6. Analyses of TNBC Cell Drug Response Data in the Genomics of Drug Sensitivity in Cancer Database

The TNBC cell line drug response data (GDSC1-raw-data) were from the Genomics of Drug Sensitivity in Cancer website (https://www.cancerrxgene.org/downloads/bulk_download). The raw viability data were normalized to the controls as relative viability. The Hill analysis and biphasic analysis were carried out as described in Section 4.3. 

## 5. Conclusions

Targeted cancer therapy is highly effective for mono-driver cancers, but effective drug combination therapy is necessary for multi-driver cancers. We previously developed a biphasic mathematical model for analyzing multi-driver cancer cell response to targeted therapy. The biphasic analysis can delineate the response of a multi-driver cancer cell to targeted therapy into a target-specific inhibition and an off-target inhibition, and predict highly potent and synergistic drug combinations for colorectal cancer cells. In the current study, we demonstrate that the same approach can be applied to triple negative breast cancer cells to identify effective targeted drug combinations for TNBC cells. 

MDA-MB-231 and MDA-MB-468 are two commonly used triple negative breast cancer cell lines. Despite numerous reports of potential protein kinase signaling pathways in sustaining their viability, a systematic understanding is lacking. Biphasic analysis of the response of these cells to a panel of protein kinase inhibitors revealed that MDA-MB-231 is Mek kinase signaling and Src kinase for viability and MDA-MB-468 is dependent on EGFR and Akt signaling for viability. Due to this multi-driver set up in these cells, individual kinase inhibitors can only partially inhibit cell viability in a target-specific manner. Higher level inhibition by a given inhibitor can only be achieved with the aid of off-target toxicity, which is not transferable to clinical efficacy in targeted therapy. The biphasic analysis further predicted and experiments confirmed effective and synergistic combination of drugs, each blocking a separate driver in each cell line. If this mechanism-based approach of developing combination targeted therapy is further validated in vivo, it would greatly expand the reach of targeted therapy to multi-driver cancers. 

## Figures and Tables

**Figure 1 cancers-12-01087-f001:**
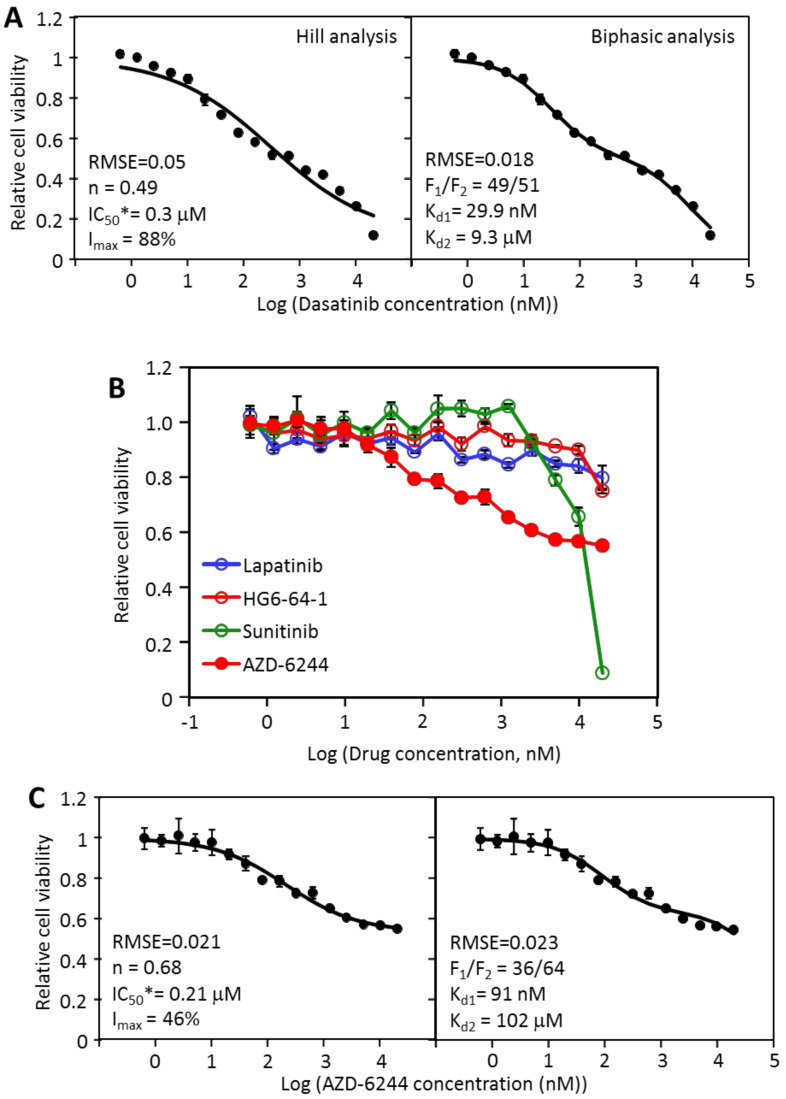
Analyses of the inhibition of MDA-MB-231 cell viability by dasatinib and AZD-6244. (**A**) the dose response data of MDA-MB-231 to dasatinib were analyzed by the Hill equation (left) and by the biphasic equation (right). The inhibitory parameters from each analysis are presented in the graphs; (**B**) inhibition of MDA-MB-231 by lapatinib, HG6-64-1, sunitinib and AZD-6244. (**C**) The dose response data of MDA-MB-231 to AZD-6244 were analyzed by the Hill equation (left) and by the biphasic equation (right). Each set of inhibition data represents six sets of data in two independent experiments in triplicates, with standard errors presented as error bars.

**Figure 2 cancers-12-01087-f002:**
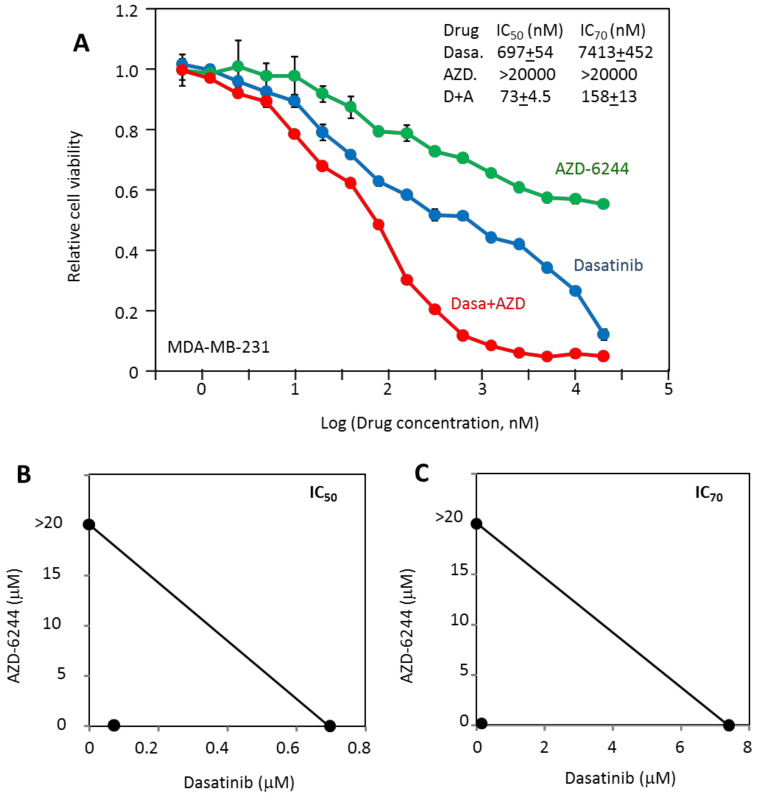
Inhibition of MDA-MB-231 cell viability by dasatinib, AZD-6244, and their combination. (**A**) The responses of MDA-MB-231 cells to dasatinib and AZD-6244 are compared to their response to the combination of both at equal molar concentrations; (**B**) isobolograms of AZD-6244 and dasatinib combination analysis at 50% inhibition; (**C**) isobolograms of AZD-6244 and dasatinib combination analysis at 70% inhibition. The results are representative of two independent experiments of triplicates.

**Figure 3 cancers-12-01087-f003:**
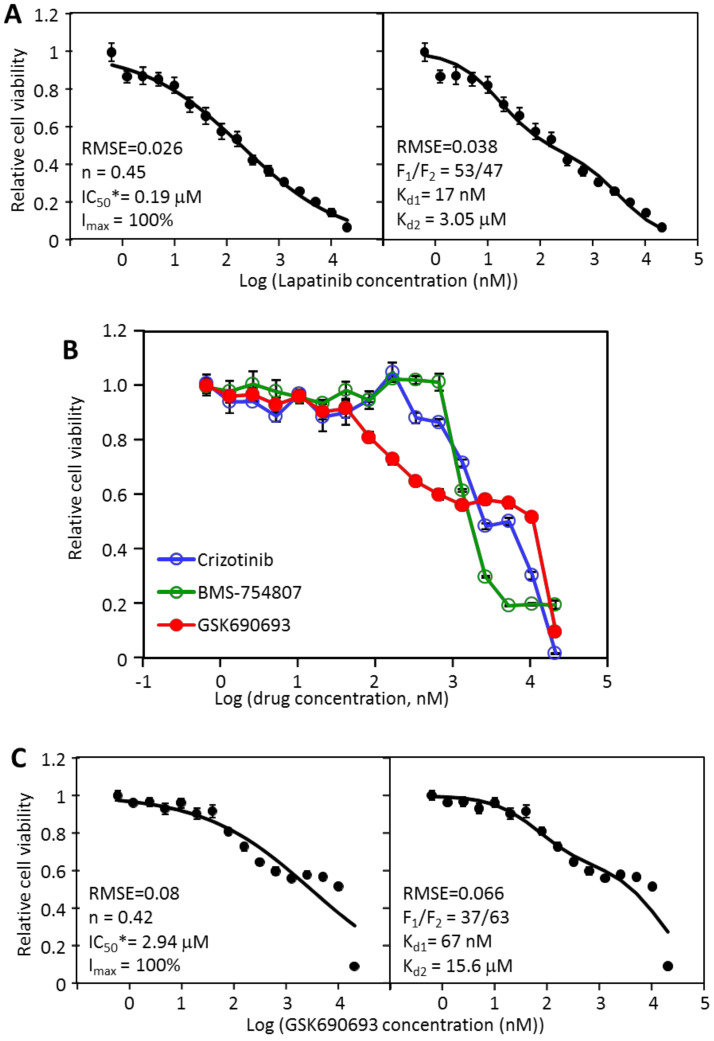
Analyses of the inhibition of MDA-MB-468 cell viability by lapatinib and GSK690693. (**A**) The dose response data of MDA-MB-468 to lapatinib were analyzed by the Hill equation (left) and by the biphasic equation (right). The inhibitory parameters from each analysis are presented in the graphs; (**B**) inhibition profile of MDA-MB-468 by crizotinib, BMS-754807, and GSK690693; (**C**) The dose response data of MDA-MB-468 to GSK690693 was analyzed by the Hill equation (left) and by the biphasic equation (right). Each set of inhibition data represents six repeat data sets from two independent experiments in triplicates, with standard errors presented as error bars.

**Figure 4 cancers-12-01087-f004:**
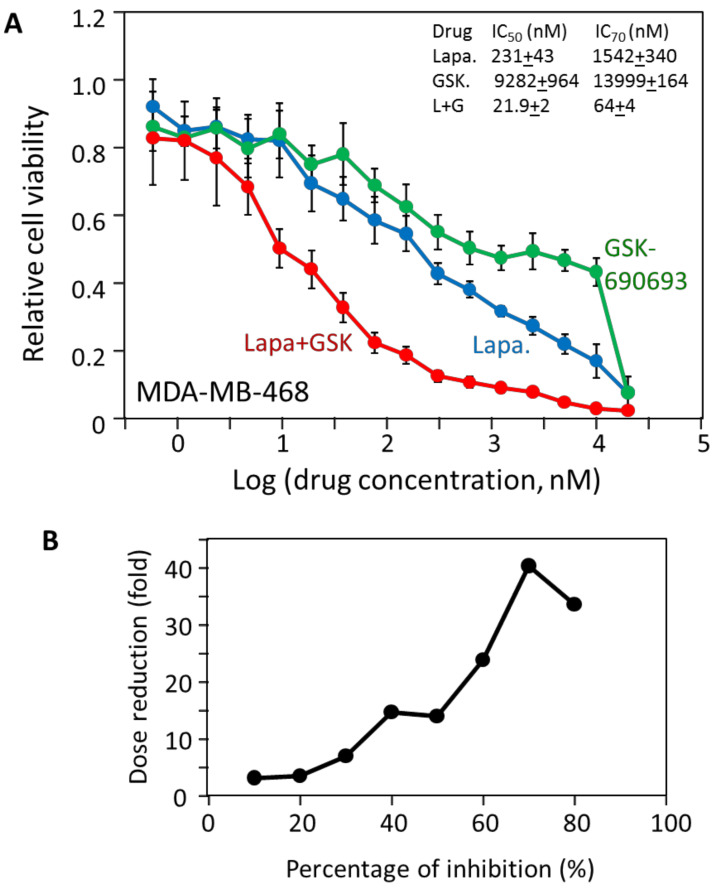
Synergistic inhibition of MDA-MB-468 cell viability by lapatinib and GSK690693. (**A**) The individual responses of MDA-MB-468 cells to lapatinib and GSK690693 are compared to the response of MDA-MB-468 cells to the combination of both drugs at equal molar concentrations; (**B**) synergistic dose reduction of lapatinib and GSK690693 on MDA-MB-468 cells as a function of the percentage of viability inhibition. The results are representative of two independent experiments of triplicates.

**Figure 5 cancers-12-01087-f005:**
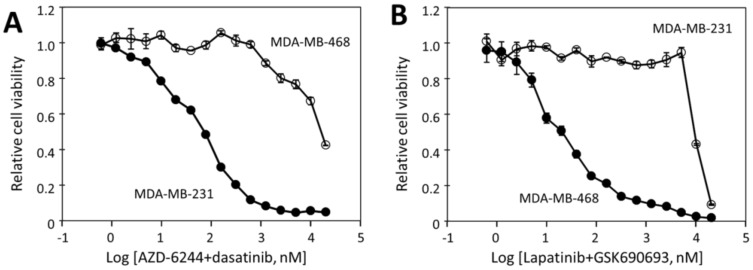
Cell line-specific inhibition of MDA-MB231 cell viability by dasatinib/AZD-6244 combination and of MDA-MB-468 cell viability by lapatinib/GSK690693 combination. (**A**) inhibition of MDA-MB-231 and MDA-MB-468 cell viability by the combination of AZD-6244 and dasatinib; (**B**) inhibition of the two cell lines by the combination of lapatinib and GSK690693. The results are representative of two independent experiments of triplicates.

**Figure 6 cancers-12-01087-f006:**
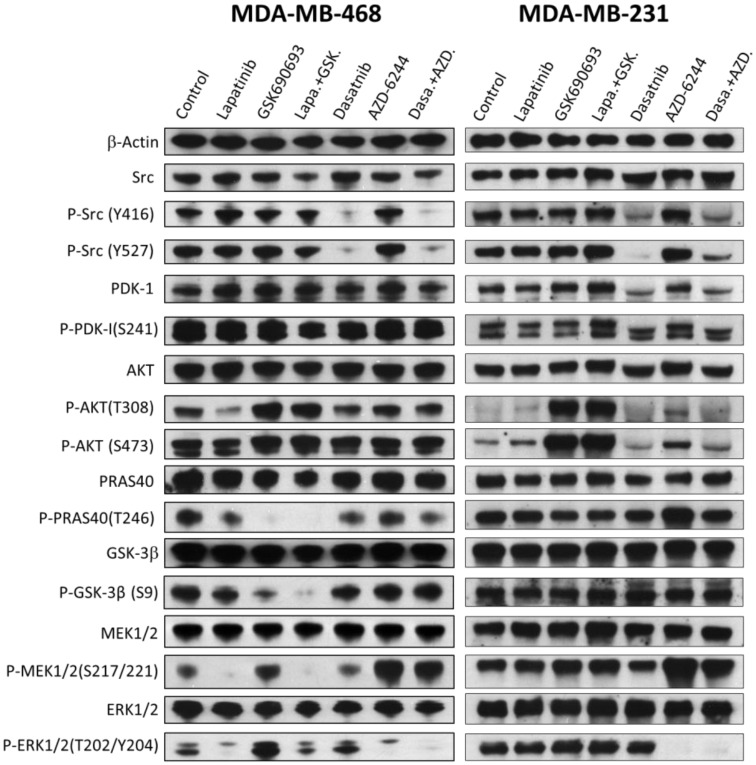
Western blot analyses of the effects of protein kinase inhibitors on the status of key signaling pathways in MDA-MB-231 and MDA-MB-468. The cells at 70% confluency were treated for 2 h with the following drugs: Control, no drug; lapatinib, 5 μM; GSK690693, 1 μM; lapatinib+GSK690693, 0.5 μM each; dasatinib, 1 μM; AZD-6244, 20 μM; dasatinib+AZD-6244, 0.5 μM each. The cells were lysed immediately in lysis SDS-PAGE sample buffer at 95 °C, and the proteins in the treated cells were fractionated by SDS-PAGE and probed by different antibodies against the indicated proteins. Protein sample loadings were equalized based on the level of β-actin. Full pictures of the Western blots and the densitometry scans are presented in Appendix A and Appendix A.

**Table 1 cancers-12-01087-t001:** Inhibition of MDA-MB-231 and MDA-MB-468 cell viability by protein kinase inhibitors.

Drug	Target ^1^	IC_50_ (μM) ^2^	
		MDA-MB-231	MDA-MB-468
Dasatinib	Src (0.2), Abl (0.05), PDGFRs (1)	0.578 + 0.05	19.7 + 0.30
AZD-6244	Mek (99)	>20	>20
Lapatinib	EGFR (2.4) ErbB2 (7)	>20	0.30 + 0.06
GSK690693	Akt (2–3)	16.7 + 0.40	9.28 + 0.96
BX-912	PDK1 (26)	14.6 + 0.63	1.62 + 0.037
AZD6482	PI 3-Kβ (10)	>20	>20
HG6-64-1	BRAF (90)	>20	>20
Bosutinib	Src (1), Abl (0.12), etc	10.3 + 0.21	>20
BMS-754807	IR (1.7), IGF-1R (1.8), Met (5.6)	7.47 + 0.15	1.59 + 0.021
Crizotinib	Alk (3.3), Met (2.1)	5.63 + 0.41	2.39 + 0.11
Erlotinib	EGFR (0.7)	>20	15.9 + 0.63
Linsitinib	IR (75), IGF-1R (35)	>20	>20
Linifanib	PDGFRs (0.6–10), VEGFRs (7–8)	>20	6.29 + 0.46
Sorafenib	PDGFRs (13), VEGFRs (31)	9.20 + 0.25	7.12 + 0.14
Sunitinib	PDGFRs (0.1–2), VEGFRs (1–2)	12.1 + 0.33	15.5 + 0.35
Pazopanib	PDGFRs (2-8), VEGFRs (14)	>20	>20
Gefitinib	EGFR (1)	>20	19.1 + 0.86
BGJ398	FGFRs (0.9–1.4)	>20	13.3 + 0.27

^1^ The table only lists common targets of these drugs and listed the K_d_’s in nM for those targets in parentheses following targets. Quantitative binding information of each drug for most of the kinome is available at the Library of Integrated Network-based Cellular Signatures (http://lincs.hms.harvard.edu/kinomescan/). ^2^ The IC_50_ values are determined empirically from the drug response curves as the concentration that resulted in 50% inhibition. Means ± SE are reported for six sets of data from two independent experiments in triplicate. The data for generating the table are presented in Appendix A for Table 1.

**Table 2 cancers-12-01087-t002:** Analysis of cancer cell response to kinase inhibitors by the Hill equation and the biphasic mathematical model. The data used for the analyses are derived from the current study and two previous studies [12,47], and the data are provided in Appendix A for Table 2. An example of Hill and biphasic analyses (the effect of dasatinib on MDA-MB-231) is shown in Appendix A—Hill and Biphasic Analyses of the Effects of Dasatinib on MDA-MB-231.

Cell	Drug	Hill Analysis				Biphasic Analysis			
		RMSE	I_max_ (%)	IC_50_* (nM)	n	RMSE	F_1_/F_2_	K_d1_ (nM)	K_d2_ (μM)
HT-29	AZD-6244	0.027	73	248	0.52	0.029	55/45	92	54.1
	BMS-754807	0.023	100	585	0.34	0.047	44/56	12	8.4
	Dasatinib	0.032	67	147	0.62	0.026	51/49	59	33.3
	HG6-64-1	0.024	51	16	0.84	0.026	50/50	14	>100
SK-CO-1	AZD-6244	0.025	83	543	0.69	0.025	55/45	169	13.6
	BMS-754807	0.034	68	200	0.5	0.048	44/56	32	29.3
NCI-H747	AZD-6244	0.019	75	71	0.62	0.029	61/39	34	25.6
	BMS-754807	0.028	75	231	0.6	0.031	54/46	75	22.4
MDA-MB-231	AZD-6244	0.021	46	210	0.68	0.023	36/64	91	>100
	Dasatinib	0.05	88	300	0.49	0.018	49/51	30	9.3
MDA-MB-468	GSK690693	0.08	100	2940	0.42	0.06	37/63	67	15.6
	Lapatinib	0.026	100	190	0.45	0.038	53/47	17	3.1
HCC-827	Gefitinib	0.027	89	10	1.73	0.056	Aug-92	11	>100
	Erlotinib	0.029	90	14	1.87	0.062	Jul-93	16	>100
	Dasatinib	0.049	93	149	1.88	0.075	Feb-98	157	>100
CTV-1	Bosutinib	0.033	96	51	2.71	0.1	100/0	56	None
	Dasatinib	0.043	97	9.1	2.72	0.11	100/0	9.1	None
	WH-4-023	0.043	98	712	2.49	0.096	100/0	716	None

**Table 3 cancers-12-01087-t003:** Analysis of DU-4475 cell line responses to BRAF, Mek, and Erk kinase inhibitors by the Hill equation and the biphasic mathematical model. The data used for the analyses are from the Genomics of Drug Sensitivity in Cancers database (Appendix A for Table 3).

Drug	Target	Hill Analysis				Biphasic Analysis			
		RMSE	I_max_ (%)	IC_50_ (nM)	n	RMSE	F_1_/F_2_	K_d1_ (nM)	K_d2_ (μM)
AZ628	BRAF	0.011	91	27	1.81	0.055	95/5	26	3469
PLX4720	BRAF	0.035	90	378	1.41	0.052	98/2	472	32448
SB590885	BRAF	0.081	75	353	1.47	0.088	86/14	524	60
RDEA119	Mek	0.030	86	37	1.62	0.055	90/10	37	75644
CI-1040	Mek	0.024	88	116	1.63	0.054	93/7	131	50525
PD-0325901	Mek	0.044	81	2.4	2.30	0.094	87/13	2.6	2169
VX-11e	Erk	0.029	98	80	1.36	0.049	100/0	80	None

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
