# Peer review of "Application of a Biphasic Mathematical Model of Cancer Cell Drug Response for Formulating Potent and Synergistic Targeted Drug Combinations to Triple Negative Breast Cancer Cells"

_cancers, 2020, doi:10.3390/cancers12051087_

Round 1

Reviewer 1 Report

This is an interesting and well written article in which the authors demonstrate, similarly as in CRC (recently published by the same group), TNBC too fits in the biphasic mathematical model to evaluate the cells response to targeted drugs. Because TNBC, like CRC, tends to be dependent on multiple drivers, the inhibition of an individual driver (by targeted therapy) causes a biphasic response that, if not conveniently analysed, can mask the potential of the drug and even the efficacy of combined therapy. Thus, they propose that, by using this biphasic mathematical model, such can possibly be overcome.

In the introduction the problem is well exposed and structured, however, in the paragraph 68-82, in which the mathematical model is explained, I felt the need to read the previous paper of the group in CRC (ref 12) to understand the model.

The rationale behind the results is easy to understand and well put. Throughout the various subsections, the authors explain the results to the reader, helping the fully comprehension of the same. I just have some remarks:

Line 139: “Because the cells did not respond to AbI inhibitor, imatinib…”, In table 1 there is no information about how the cell line MDA-MB-321 responds to imatinib, so there are no results to sustain this claim as it is.

Line 202-203: “ …with an IC50 and IC70 of 231nM and 1542nM, respectively (Table 1).” Table 1 is referred, however in the latter there is no information regarding the IC70 of the cell line MDA-MB-468. If not in table 1, the missing information could be indexed to the supplementary material.  

Regarding the discussion and conclusions, it is well structured and adequate. Even though in vivo testing is needed, they justify well the potential of further studying the application of their mathematical model in the future, with the final purpose of providing the most suitable analysis to multi-driver cancers, enabling the discovery of new combined targeted drug strategies.

Author Response

Comment 1: This is an interesting and well written article in which the authors demonstrate, similarly as in CRC (recently published by the same group), TNBC too fits in the biphasic mathematical model to evaluate the cells response to targeted drugs. Because TNBC, like CRC, tends to be dependent on multiple drivers, the inhibition of an individual driver (by targeted therapy) causes a biphasic response that, if not conveniently analysed, can mask the potential of the drug and even the efficacy of combined therapy. Thus, they propose that, by using this biphasic mathematical model, such can possibly be overcome.

Response: Thank you for this comment.

Comment 2: In the introduction the problem is well exposed and structured, however, in the paragraph 68-82, in which the mathematical model is explained, I felt the need to read the previous paper of the group in CRC (ref 12) to understand the model.

Response: Thank you for pointing out this concern. We added the equation and a description to this discussion (Lines 77-79). This revision should make the presentation easier to follow. If a reader likes to get the details of the model in application to colorectal cancer cells, it would still be necessary to read the previous paper.

Comment 3: Line 139: “Because the cells did not respond to AbI inhibitor, imatinib…”, In table 1 there is no information about how the cell line MDA-MB-321 responds to imatinib, so there are no results to sustain this claim as it is.

Response: The reviewer is correct in pointing this out. Imatinib was indeed not included in the data shown in Table 1. In early screening, we tested compounds at four concentrations, 10 nM, 100 nM, 1 microM and 10 microM. Imatinib, up to 10 microM, did not significantly inhibit MDA-MB-231 or other cell lines we tested, and we dropped this compound from later refined testing. The results for imatinib on MDA-MB-231 cells, tested twice in quadruplicates, are shown in the following table:

Concentration

0 nM

10 nM

100 nM

1 microM

10 microM

Cell viability (%)

100

100.13

103.29

101.36

95.92

SE

0

1.60

3.32

2.86

2.98

We put this data and other data for Table 1 in Supplementary Data for Table 1, and inserted a reference to it in line 143: "did not respond to Abl inhibitor, imatinib (Supplementary Data for Table 1), ".

Comment 4: Line 202-203: “ …with an IC50 and IC70 of 231nM and 1542nM, respectively (Table 1).” Table 1 is referred, however in the latter there is no information regarding the IC70 of the cell line MDA-MB-468. If not in table 1, the missing information could be indexed to the supplementary material.  

Response: We thank the reviewer for pointing this out. We removed the reference to IC70 in this context (lines 207-208). When it is discussed in later contexts, it is clearly visible from the graphs in discussion.

Comment 5: Regarding the discussion and conclusions, it is well structured and adequate. Even though in vivo testing is needed, they justify well the potential of further studying the application of their mathematical model in the future, with the final purpose of providing the most suitable analysis to multi-driver cancers, enabling the discovery of new combined targeted drug strategies.

Response: We thank the reviewer for this comment.

Reviewer 2 Report

Shen et al provide a thorough analysis of how drug response is able to differentiate between single and multi-driver TNBC cell lines. The manuscript is well presented and the conclusions are supported by the experiments and properly toned down.

I have one problem regarding the manuscript: the authors used only 1:1 combination of the two drugs when they constructed their isobolograms. The concentrations were not normalised according to the IC50 value of the two drugs, which I think would be much more meaningful.  Due to this, the drugs synergy and how they affect the cells through the different concentrations are not visible. I suggest additional experiments to test the drugs in different ratio of their IC50 concentration.

Another question which I have regarding Figure 3A and general: How can be sure that there is no additional partial very low concentration inhibition in of the molecules. Why are we sure that there is only a biphasic drug response curve and not let's say three or more? How can we determine it? Maybe it worth a few sentences in the discussion.

Author Response

Comment 1: Shen et al provide a thorough analysis of how drug response is able to differentiate between single and multi-driver TNBC cell lines. The manuscript is well presented and the conclusions are supported by the experiments and properly toned down.

Response: We appreciate this comment.

Comment 2: I have one problem regarding the manuscript: the authors used only 1:1 combination of the two drugs when they constructed their isobolograms. The concentrations were not normalised according to the IC50 value of the two drugs, which I think would be much more meaningful.  Due to this, the drugs synergy and how they affect the cells through the different concentrations are not visible. I suggest additional experiments to test the drugs in different ratio of their IC50 concentration.

Response: We thank the reviewer for raising this important point. We considered varying the ratio according to their IC50's, but decided not to because the IC50's did not reflect the target-specific inhibition. Two of the drugs, AZD-6244 for MDA-MB-231 and GSK690693 for MDA-MB-468, did not reach 50% inhibition up to 20 mM. We also considered varying the ratio according to Kd1's, but the Kd1's all generally fall in the same ballpark in the double-digit nM range, varying the ratios according to the individual Kd1's wouldn't change the ratio too much, so we decided to stick to the 1:1 ratio. We agree that there will be situations where varying the ratio may derive more significant synergistic benefits, and will incorporate it into our future experimental design.

Comment 3: Another question which I have regarding Figure 3A and general: How can be sure that there is no additional partial very low concentration inhibition in of the molecules. Why are we sure that there is only a biphasic drug response curve and not let's say three or more? How can we determine it? Maybe it worth a few sentences in the discussion.

Response 3: This is an excellent point. We added the following paragraph to the discussion: "Theoretically, the inhibition pattern may contain more than two phases. For example, if a cell contains three drivers and each responsible for 33% cell viability, a drug that inhibits driver one with a Kd of 10 nM, the driver two with a Kd of 100 nM, and causes off-target toxicity with a Kd of 5 microM, then the inhibition will be theoretically triphasic. Such perfectly configured scenarios are likely rare. In practice, if two drivers both respond to the same inhibitor with Kd's within 10 fold of each other, they become indistinguishable by this type of analysis. The toxicity phase is also likely complex as it is caused by ill-defined and likely variable mechanisms. Despite these potentially complicating scenarios, the inhibition of most multi-driver cancer cell by most inhibitors for one of the drivers can be practically described by this biphasic mathematical model." (line 496-504).

Reviewer 3 Report

In this article, the authors test the applicability of their previously described biphasic mathematical model on two triple negative breast cancers lines for characterizing the mechanism of cell proliferation i.e. multi driver vs mono-driver hypothesis. They further identify the potential drug combinations that can independently and effectively target the identified drivers at concentrations that are presumably not expected to cause general toxicity. The analysis is also useful in revealing observations such as AZD-6244 does not reach 50% inhibition at 20 μM but can cause partial inhibition even at lower nM concentrations. I understand, this knowledge could be useful especially when formulating drug combinations against multi-driver cancers.

The manuscript is nicely constructed and the conclusions are well supported by experiments that includes validation by cell viability assays and western blotting analysis. However, I would like to hear authors thoughts on toxic effects from drug combinations when compared to that expected from individual application of drugs i.e. would the drug when given in combinations show synergy in toxic effects?

Author Response

Comment 1: In this article, the authors test the applicability of their previously described biphasic mathematical model on two triple negative breast cancers lines for characterizing the mechanism of cell proliferation i.e. multi driver vs mono-driver hypothesis. They further identify the potential drug combinations that can independently and effectively target the identified drivers at concentrations that are presumably not expected to cause general toxicity. The analysis is also useful in revealing observations such as AZD-6244 does not reach 50% inhibition at 20 μM but can cause partial inhibition even at lower nM concentrations. I understand, this knowledge could be useful especially when formulating drug combinations against multi-driver cancers.

Response 1: We appreciate these positive comments.

Comment 2: The manuscript is nicely constructed and the conclusions are well supported by experiments that include validation by cell viability assays and western blotting analysis. However, I would like to hear authors thoughts on toxic effects from drug combinations when compared to that expected from individual application of drugs i.e. would the drug when given in combinations show synergy in toxic effects?

Response 2: We appreciate this excellent suggestion and added the following paragraph to discuss this question (lines 466-477): "Even though the target-specific effects are combined synergistically when appropriate drugs are used together, data in this study and a previous study [12] suggest that the off-target effects of the individual drugs are not combined synergistically. For example, dasatinib and AZD-6244 individually inhibited MDA-MB-468 with IC50's of around 20 microM or above (Table 1), and the combination inhibited the cell line with IC50 close to 20 microM of each (Fig. 5A). Similarly, lapatinib and GSK690693 inhibited MDA-MB-231 with IC50's of >20 microM and 16.7 microM, respectively (Table 1), and the combination inhibited the cell line viability with an IC50 close to 10 microM of each (Fig. 5B). Similar observations were made with drugs that inhibited HT-29 cells with only off-target effects [12]. A broader question about this combination approach is whether these cell line-specific drug combinations would display any potent general toxicity toward any normal cells. Drug combinations should be tested against a broad spectrum of normal cells and/or animal models to assess their toxicity before they can be used in clinical setting."

Reviewer 4 Report

The manuscript by Shen and colleagues describes the application of a biphasic model to assess the effect of small molecule kinase inhibitors on cancer cell viability. Using a biphasic model to analyze the effect of inhibiting a given protein kinase in cell viability, the authors show, that triple negative breast cancer (TNBC) is a multi-driver disease, and speculate that kinase inhibitors might affect viability through on-target and off-target effects. While this work advances our understanding of kinase inhibitors mechanism of action, the fact that kinase inhibitors display off-target effects is widely known in the field. In addition, it is not clear whether this work represents an advance from the author's previous publication (i.e. Shen et al. 2020), rather than using a different cancer model. As stated by the authors, the work would greatly benefit from testing synergistic combinations in vivo, however, this reviewer appreciates it is out of the scope of the current work.

The authors need to address the concerns listed below.

MAJOR

  1. The authors must provide the data they used to generate Tables 1, 2 and 3 as supplemental figures.
  2. The authors display data from growth inhibition curves representing 6 data sets from two independent experiments. In this reviewer's opinion, this is not the right way to analyze these experiments. The authors should analyze each independent experiment by itself, and then calculate the average -/+ from both experiments.
  3. The authors need to provide full scans of their western blots and include molecular weight markers, both in Fig. 6 and in supplemental data.
  4. It is not stated if changes in downstream signaling were analyzed once or n>1.
  5. In addition, it is likely the authors needed to run more than one gel to analyze specific phosphorylated and total protein levels. Therefore, each signal should be normalized to the corresponding gel loading control. It would be more accurate if the authors provide p-protein signal/total protein signal ratios.
  6. Lines 329-332: it is widely described that treatment with AKT inhibitors result in an increase of AKT phosphorylation and a reduction of its activity (e.g. Davies et al. Mol. Cancer Ther., 2012; 10.1158/1535-7163.MCT-11-0824-T). The authors need to re-consider their interpretation of their data. 

MINOR

  1. Line 13: the authors should clarify that TNBC is defined as lacking ERBB2 amplification.
  2. Line 54: the authors should correct the gene PIK2CA by PIK3CA.
  3. Line 56: genes should be written in italics.
  4. Line 58: the authors need to indicate what VEGFR stands for.
  5. Table 1: include IC50 each inhibitor IC50 for the target it was designed against.
  6. In all figure legends the authors should indicate inhibition of (cell line name) viability, as this is the effect they are measuring.
  7. Include in Fig 4A calculation of different parameters as in Fig 2A.
  8. Check MDA-MB-468 for PTEN expression.
  9. Table 2: include the target/s against which each inhibitor was designed.
  10. Do pGSK3b S9 or pPRAS40 T246 diminish in MDA-MB-231 after treatment with other AKT inhibitor, e.g. AZD5363 or MK-2206?
  11. In the methods section, the authors state they seeded 25,000 cells/well in 96-well plate for viability. This means cells were above 10,000cells/cm2 resulting in very high cell confluence that might inhibit proliferation.
  12. Very high cell confluence can affect analysis of signaling pathways, the authors should analyze the effect of inhibitors in exponentially growing cells (i.e. 50-60% confluence).

Author Response

We greatly appreciate the many excellent suggestions from this reviewer.

Comment 1: The manuscript by Shen and colleagues describes the application of a biphasic model to assess the effect of small molecule kinase inhibitors on cancer cell viability. Using a biphasic model to analyze the effect of inhibiting a given protein kinase in cell viability, the authors show, that triple negative breast cancer (TNBC) is a multi-driver disease, and speculate that kinase inhibitors might affect viability through on-target and off-target effects. While this work advances our understanding of kinase inhibitors mechanism of action, the fact that kinase inhibitors display off-target effects is widely known in the field. In addition, it is not clear whether this work represents an advance from the author's previous publication (i.e. Shen et al. 2020), rather than using a different cancer model. As stated by the authors, the work would greatly benefit from testing synergistic combinations in vivo, however, this reviewer appreciates it is out of the scope of the current work.

Response: There are two concerns in this comment. 1) ---the fact that kinase inhibitors display off-target effects is widely known in the field. The reviewer is correct that the off-target effects by kinase inhibitors are well-known, but how to evaluate target-specific versus off-target effects of a given inhibitor is not well studied. Most studies use the IC50 value as a measure of inhibitory potency, which is the combined inhibition through both target-specific and off-target effects. We demonstrate that delineating the contribution of these two effects is of fundamental importance for formulating effective combination targeted therapy. 2) ---it is not clear whether this work represents an advance from the author's previous publication (i.e. Shen et al. 2020), rather than using a different cancer model. We understand this concern, but argue that this is not a simple repeat of the previous study in different cells. The biphasic inhibition model for multi-driver cancers is useful only if it is broadly applicable beyond a few selected cell lines. Extension to different cancer type with distinct drug combinations through the current study is an important step in establishing potential broad applicability of this approach. Many aspects of this model still wait to be defined, and we hope that these two studies will make a convincing case that this approach of generating effective combination targeted therapy is worth further investigation toward a broader spectrum of cancer cells.

Major Comment 1: The authors must provide the data they used to generate Tables 1, 2 and 3 as supplemental figures.

Response: The data used for generating these tables are provided as Supplementary Data for Table 1, Supplementary Data for Table 2, and Supplementary Data for Table 3. They are referenced in appropriate contexts.

Major Comment 2: The authors display data from growth inhibition curves representing 6 data sets from two independent experiments. In this reviewer's opinion, this is not the right way to analyze these experiments. The authors should analyze each independent experiment by itself, and then calculate the average -/+ from both experiments.

Response: We considered both options of analyzing and presenting the data, and chose the option we used. This option de-emphasizes the differences between the two experiments and puts the emphasis on what the data collectively reveal. Both options yield the same conclusions, and the option we used treats all data equally. The same approach was used in the previous paper analyzing the colorectal cancer cell data. For the sake of consistency with the previous paper, we prefer using the same approach.

Major Comment 3: The authors need to provide full scans of their western blots and include molecular weight markers, both in Fig. 6 and in supplemental data.

Response: The pictures provided in Supplementary Fig. S1 are full images of the Western blots. We ran separate gels and western blots for each antibody for each set of samples, and overwhelming majority of the time, there is only one band in each lane. Each image in Fig. S1 represents an individual Western blot. The molecular weight markers were on the Western blot membranes and we determined the size and identity of the Western blot bands by carefully matching the blot band to those of the protein standards on the membrane.

Major Comment 4: It is not stated if changes in downstream signaling were analyzed once or n>1.

Response: The downstream signaling was analyzed at least two times, with many of the Western blots performed more than two times. The results from other analyses may not be as complete or as presentable as the results used in the paper, but they were consistent with what is presented in the paper. We added the note "Western blotting analyses were performed on two separate experiments." (lines 591-592).

Major Comment 5: In addition, it is likely the authors needed to run more than one gel to analyze specific phosphorylated and total protein levels. Therefore, each signal should be normalized to the corresponding gel loading control. It would be more accurate if the authors provide p-protein signal/total protein signal ratios.

Response: This is an excellent suggestion. Separate tables showing the p-protein signal/total protein signal ratios are provided in Supplementary Table 1.  

Major Comment 6: Lines 329-332: it is widely described that treatment with AKT inhibitors result in an increase of AKT phosphorylation and a reduction of its activity (e.g. Davies et al. Mol. Cancer Ther., 2012; 10.1158/1535-7163.MCT-11-0824-T). The authors need to re-consider their interpretation of their data. 

Response: We thank the reviewer for making this excellent suggestion. We revised the relevant section to the following, with additional reference (lines 341-346): "While the observation that Akt phosphorylation on both T308 and S473 was increased by GSK690693 was consistent with earlier observations with multiple cell lines and multiple Akt inhibitors [12,23,31-33], the fact that GSK690693 and GSK690693/lapatinib combination did not inhibit the phosphorylation of PRAS40 and GSK-3beta explains why MDA-MB-231 was not inhibited by GSK690693 or the combination. It is not yet clear why GSK690693 did not inhibit Akt phosphorylation of PRAS40 and GSK-3beta in MDA-MB-231 cells."

Minor Comment 1: Line 13: the authors should clarify that TNBC is defined as lacking ERBB2 amplification.

Response: We modified the said sentence to (lines 13-14): "Triple negative breast cancer is a collection of heterogeneous breast cancers that are immunohistochemically negative for estrogen receptor, progesterone receptor and ErbB2 (due to deletion or lack of amplification)." The underlined phrases are newly inserted to better define TNBC regarding ErbB2.

Minor Comment 2: Line 54: the authors should correct the gene PIK2CA by PIK3CA.

Response: Corrected (line 54). Thank you.

Minor Comment 3: Line 56: genes should be written in italics.

Response: Corrected (line 56). Thank you.

Minor Comment 4: Line 58: the authors need to indicate what VEGFR stands for.

Response: Added (lines 58-59). Thank you.

Minor Comment 5: Table 1: include IC50 each inhibitor IC50 for the target it was designed against.

Response: They are now added in the parentheses following the name of the targets, which is noted in the note for the table. Thank you for the suggestion.

Minor Comment 6: In all figure legends the authors should indicate inhibition of (cell line name) viability, as this is the effect they are measuring.

Response: Added to each relevant figure and Table 1. Thank you.

Minor Comment 7: Include in Fig 4A calculation of different parameters as in Fig 2A.

Response: Added. Thank you.

Minor Comment 8: Check MDA-MB-468 for PTEN expression.

Response: Thank you for this excellent suggestion. MDA-MB-468 is indeed deficient in PTEN expression, which explains the Akt activation in this cell line and its sensitivity to GSK690693. We revised the relevant section as follows (lines 305-312): "According to the COSMIC database, MDA-MB-468 overexpresses EGFR [28,29], which explains why this cell line is sensitive to lapatinib. Furthermore, it is well established that MDA-MB-468 cells are deficient in PTEN expression [30-34], which would lead to constitutive activation of Akt signaling. This observation explains the partial reliance of MDA-MB-468 on Akt activity indicated by the cell line sensitivity to GSK690693."

Minor Comment 9: Table 2: include the target/s against which each inhibitor was designed.

Response: We appreciate this suggestion, and adding the targets for each inhibitor would make it more convenient for the reader. However, the table is already quite busy, making it hard to add a target column. The information is already given in Table 1, and we prefer not adding this information here. If the reviewer and editor feel strongly about adding the information, we can do that in further revision/galley proof.

Minor Comment 10: Do pGSK3b S9 or pPRAS40 T246 diminish in MDA-MB-231 after treatment with other AKT inhibitor, e.g. AZD5363 or MK-2206?

Response: This is an excellent question. Reports in the literature indicate that MK-2206 does not inhibit MDA-MB-231 cell viability up to 12 mM, even though MK-2206 does inhibit Akt phosphorylation. We did not find any reference to whether MK-2206 inhibits the phosphorylation of PRAS40 and GSK3beta in MDA-MB-231 cells. We incorporated this information in the text with additional references. We added the following sentence: "It is not yet clear why GSK690693 did not inhibit Akt phosphorylation of PRAS40 and GSK-3beta in MDA-MB-231 cells. These results are consistent with literature reports that another Akt inhibitor, MK-2206 [39-41], also does not inhibit MDA-MB-231 cell viability, even though it inhibits Akt phosprylation in MDA-MB-231 cells." (lines 345-349).

Minor Comment 11: In the methods section, the authors state they seeded 25,000 cells/well in 96-well plate for viability. This means cells were above 10,000cells/cm2 resulting in very high cell confluence that might inhibit proliferation.

Response: According to the guideline of ATCC, MDA-MB-231 and MDA-MB-468 cells should be seeded at 3-5x104/cm2, and there are about 2x105 cells/cm2 at 80-90% confluency. The doubling time is 38h for MDA-MB-231 and 47h for MDA-MB-468. Our seeding level is around 25% confluency and the control wells reach 60-75% confluency at the end of 72-hour incubation. This seeding level was also optimized experimentally: seeding at much lower levels increases signal noise, and seeding at much higher levels results in growth inhibition in the control wells as suggested by the reviewer.

Minor Comment 12: Very high cell confluence can affect analysis of signaling pathways, the authors should analyze the effect of inhibitors in exponentially growing cells (i.e. 50-60% confluence).

Response: For analyzing the drug effects on signaling pathways, we originally stated that the cells were cultures to 70% confluency, but in reality it is a range of 60% to 70%. The cells were treated by the drugs for 2 h, so additional growth is negligible. We revised the "70%" to "60-70%" (line 585).

Round 2

Reviewer 2 Report

The authors addressed my comments and explained why they do not use a different concentration. The work by itself is really useful for determining the mono-multi driver situation and points towards potential drug synergies, but not measures it explicitly.

I can stand behind that. Good luck!

Author Response

Thank you for the positive comments.

Reviewer 4 Report

I appreciate the authors have addressed most of my concerns and their manuscript has improved. I still have a couple of concerns before the acceptance of this work for publication.

Major

1. The authors did not address my comment #3 from the previous revision: "The authors need to provide full scans of their western blots and include molecular weight markers, both in Fig. 6 and in supplemental data." They still need to show the original raw images that should include molecular weights and indicate where the authors cropped to generate the main figure.

Minor

1. The authors should plot data included in supplementary tables 1-3 as to make it visible how they calculated the reported IC50.

2. Lines 343-345 read: "These results are consistent with literature reports that another Akt inhibitor, MK-2206 [39-41], also does not inhibit MDA-MB-231 cell viability, even though it inhibits Akt phosprylation in MDA-MB-231 cells." The authors show that the AKT inhibitor GSK690693 increases AKT phosphorylation in MDA-MB-231 (suggesting it is targeting/binding to AKT). This could be indicated. There is a typo that needs to be corrected.

Author Response

Major Comment 1: The authors did not address my comment #3 from the previous revision: "The authors need to provide full scans of their western blots and include molecular weight markers, both in Fig. 6 and in supplemental data." They still need to show the original raw images that should include molecular weights and indicate where the authors cropped to generate the main figure.

Response: We expanded the images in Supplementary Fig. S1 to include the whole length of the blots, and labeled the images with molecular weight markers. As it is clear from examining these images, overwhelming majority of the images contain only one band in each lane. In cases there are additional bands, either due to nonspecific recognition or degradation products, the identity of the intended protein in each case was clear-cut based on the expected size and the molecular weight markers. In each of the full images, we used dashed lines to indicate cropping borders to generate the images used in Fig. 6.

Minor Comment 1: The authors should plot data included in supplementary tables 1-3 as to make it visible how they calculated the reported IC50.

Response: Supplementary tables 1 through 3 deal with different types of data and processing, and below is how the data are processed for each table.

Table 1 and Supplementary Data for Table 1. Table 1 in the manuscript reports the IC50 for each drug against each cell line. Each of the IC50 value and the SE are calculated from the individual IC50 values manually determined from the individual curves. For example, the IC50 of dasatinib against MDA-MB-231 is calculated from six IC50 values of 0.40738, 0.616595, 0.562341, 0.724436, 0.446684, and 0.707946, giving an average of 0.578 mM and SE of 0.054 mM. Each of these individual IC50 values was manually determined from the drug response curves. For other researchers interested in the IC50 values this table provides sufficient information. The values reported in the Table S1 are the average cell viability and SE at each drug concentration. For other researchers who may be interested in the dose response curves, the Supplementary Data provide detailed dose response data.

Table 2 and Supplementary Data for Table 2. Table 2 in the manuscript reports the results of Hill analysis and biphasic analysis. The equations used for the analyses are described in detail in the manuscript. Table S2 reports the data (collected from data in this paper and previously published data) used for the analyses. When the data in Table S2 are plugged in the equations, the parameters reported in Table 2 will be obtained. We are adding a Microsoft Excel workbook as "Supplementary Data – Hill and Biphasic Analyses of the Effects of Dasatinib on MDA-MB-231", showing an example of these calculations. The calculations in this worksheet turn the Supplementary Data into the parameters in Table 2.

Table 3 and Supplementary Data for Table 3. The relationship between Table 3 and the Supplementary data for Table 3 is the same as that between Table 2 and the Supplementary Data for Table 2. The analysis is identical to that shown in the example of dasatinib on MDA-MB-231 cells. The data in Table S3 can be fed into the biphasic analysis and Hill analysis to obtain the biphasic and Hill parameters in Table 3.

Minor Comment 2: Lines 343-345 read: "These results are consistent with literature reports that another Akt inhibitor, MK-2206 [39-41], also does not inhibit MDA-MB-231 cell viability, even though it inhibits Akt phosprylation in MDA-MB-231 cells." The authors show that the AKT inhibitor GSK690693 increases AKT phosphorylation in MDA-MB-231 (suggesting it is targeting/binding to AKT). This could be indicated. There is a typo that needs to be corrected.

Response: We appreciate the correction and the suggestion. We corrected the typo, changing "phosprylation" to "phosphorylation". We also revised the sentences in question as suggested to indicate that GSK690693 entered MDA-MB-231 cells and interacted with Akt. The revised sentences read as follows: "The observation that Akt phosphorylation on both T308 and S473 was increased by GSK690693 indicated that GSK690693 entered the cells, and interacted with Akt. Increased Akt phosphorylation in response to GSK690693 was previously observed in multiple cell lines [12,23,36-38]. The fact that GSK690693 and GSK690693/lapatinib combination did not inhibit the phosphorylation of PRAS40 and GSK-3beta explains why MDA-MB-231 was not inhibited by GSK690693 or the combination." (lines 342-346).

We appreciate all the comments from this reviewer. The revisions according to these comments helped significantly improve the manuscript. Thank you.

Round 3

Reviewer 4 Report

I thank the authors for addressing all of my concerns.